# Genome-Wide Analysis and Expression Profiling of the JAZ Gene Family in Response to Abiotic Stress in Alfalfa

**DOI:** 10.3390/ijms26104684

**Published:** 2025-05-14

**Authors:** Xiaohong Li, Aneela Bashir, Huizheng Yang, Ansar Abbas, Yaoyao Li, Xin Zeng, Longkao Zhu, Qinke Shi, Mamateliy Tursunniyaz, Lijing Zhang

**Affiliations:** 1College of Pastoral Agriculture Science and Technology, Lanzhou University, Lanzhou 730030, China; lxh0895@163.com (X.L.); aneela2024@lzu.edu.cn (A.B.); yanghzh2024@lzu.edu.cn (H.Y.); ansar2024@lzu.edu.cn (A.A.); liyaoy2023@163.com (Y.L.); zx18383835001@163.com (X.Z.); zhulk2024@lzu.edu.cn (L.Z.); shiqk2023@lzu.edu.cn (Q.S.); pojunmuyan@163.com (M.T.); 2State Key Laboratory of Herbage Improvement and Grassland Agro Ecosystems, Lanzhou 730030, China; 3Key Laboratory of Grassland Livestock Industry Innovation, Ministry of Agriculture and Rural Affairs, Lanzhou 730030, China; 4Engineering Research Center of Grassland Industry Ministry of Education, Lanzhou 730030, China

**Keywords:** alfalfa, *JAZ* gene family, genome wide, expression pattern, abiotic stress

## Abstract

The Jasmonate ZIM-domain (JAZ) proteins act as repressors in the Jasmonate (JAs) signaling pathway, and play a critical role in regulating plant growth, development, and responses to biotic and abiotic stresses. In this study, bioinformatics methods were employed to identify the *JAZ* gene family in the whole genome of alfalfa (*Medicago sativa* cv. Zhongmu No. 1) and systematically analyze their gene characteristics, subcellular localization, phylogenetic evolution, promoter *cis*-elements, expression patterns, and responses to abiotic stress. A total of nine *MsJAZ* gene family members with complete TIFY and Jas domains were identified; they were distributed unevenly across four chromosomes and encoding proteins ranging from 94 aa (MsJAZ2) to 337 aa (MsJAZ7), with molecular weights (MWs) from 19.33 to 38.03 kDa. Phylogenetic analysis showed that the *MsJAZ* gene family could be classified into four clades (Clades I–II, IV–V), which are closely related to citrus. Most *MsJAZ* family members contain light-responsive, hormone-responsive, and stress-responsive *cis*-elements. Subcellular localization results indicated that all *MsJAZ* genes are expressed and function in the nucleus. The RT-qPCR results showed that *MsJAZ* genes were primarily expressed in the leaves and petioles. Under salt and drought stress, all *MsJAZ* genes exhibited varying degrees of response, with *MsJAZ4* and *MsJAZ7* showing the most pronounced reactions. Meanwhile, under chromium (Cr) and MeJA stress, both *MsJAZ4* and *MsJAZ9* exhibited strong responses. Subcellular localization results showed that the MsJAZ4/7 protein was localized on the plasma membrane and nucleus. The yeast adversity test showed that the *MsJAZ4/7* gene was more sensitive to salt stress. This study provides a foundation for future research on the function of the *MsJAZ* genes and its regulatory mechanism, as well as for identifying candidate genes for alfalfa stress tolerance breeding.

## 1. Introduction

Alfalfa (*Medicago sativa*), a legume, is one of the first fodder crops to be domesticated and has been cultivated in more than 80 countries [1]. It is excellent leguminous forage and is known as the “queen of forages” because of its high forage yield and protein, vitamin, and mineral content [2]. Additionally, alfalfa possesses a strong root system characterized by intertwining and dense horizontal roots that create an underground protective network. This feature enhances its suitability for soil and water conservation, windbreaks, sand fixation, slope protection, and soil stabilization, thereby playing a significant role in ecological management [3,4]. However, the changing global climate environment has posed unprecedented challenges to alfalfa production, such as pests and diseases, drought, and soil salinization. Consequently, the development of new varieties that are tolerant to drought and salinity, as well as resistant to pests and diseases through molecular breeding, has emerged as the most effective approach to address this challenge.

Phytohormones regulate various stages of plant growth and development and response and adaptation to environmental factors, such as auxins (IAA), cytokinins, gibberellins (GA), ethylene (ETH), brassinosteroids (BR), jasmonates (JAs), salicylic acid (SA), and strigolactones (SL) [5]. JA is an important phytohormone in plants which plays an important role in regulating root growth, flowering, leaf senescence, fruit ripening, and growth and development [6,7,8]. This research found that JA has been implicated in plant responses to biotic stresses such as herbivory, pathogenic bacteria, and viral infection, enhancing or reducing plant tolerance to various stresses [9]. Jasmonate isoleucine (JA-Ile) is the most biologically active JA, and its co-receptor Skp1-Cullin1 F-box (SCF) protein-ubiquitin ligase complex SCFCOI1-JAZ senses JA-Ile in the nucleus to deregulate target gene transcriptional repression [10]. Jasmonic acid biosynthesis occurs in chloroplasts using cell membrane-released α-linolenic acid as a precursor. This process is catalyzed by lipoxygenase (LOX) to produce 13S-hydroperoxynolenic acid (13-HPOT). Subsequently, 13-HPOT is converted by propylene oxide synthase (AOS) and propylene oxide cyclase (AOC) to 12-oxo-phytodienoic acid (OPDA). OPDA is translocated from chloroplasts to the peroxisome, where it is catalytically reduced by reductase (OPR3) and converted to (+)-7-iso-JA by three β-oxidation reactions. (+)-7-iso-JA is then transported to the cytoplasm, where it binds to isoleucine (Ile) by adenylyl-forming enzyme (JAR1) to form the highly biologically active jasmonate–isoleucine complex (JA- Ile) [11]. JAZ proteins were nuclear zinc-finger proteins containing mainly the TIFY (also known as ZIM) and Jas (also known as CCT-2) structural domains [12]. The TIFY structural domain is the signature feature of JAZ proteins and consists of approximately 50–70 amino acid residues. The domain contains a conserved α-helix region (HePTc) that interacts with NINJA and TPL (TOPLESS) proteins and is sufficient to form larger and more complex JAZ signaling complexes, enhancing regulation of downstream gene expression [13]. Meanwhile, the N-terminal domain of JAZ proteins interacts with JA-responsive transcription factors (e.g., MYC family members). When JAZ binds to MYC2, the transcriptional activation of MYC2 is inhibited, suppressing the expression of jasmonate-responsive genes [14]. Upon degradation of JAZ, these transcription factors are released, restoring their activity and enabling the jasmonate signaling response.

JAs were phytohormones essential for plant development and environmental adaptation and are key regulators of responses to biotic and abiotic stresses [15]. Under stress or endogenous stimuli, plants accumulate bioactive JA [16]. The overexpression of *AtJAZ1* in *Arabidopsis thaliana* enhances the plant’s resistance to the beet moth (*Scrobipalpa ocellatella*) [17,18]. Overexpression of *GsJAZ2* improves salinity stress tolerance in soybean [19], and overexpression of *CmJAZ1* in chrysanthemum can regulate the flowering time of chrysanthemum [20]. *OsJAZ1* has been identified as a negative regulator of drought tolerance, primarily through its modulation of the JA and ABA signaling pathways [21]. The overexpression of the *GsJAZ2* gene (*Glycine max*) significantly enhances the salt stress tolerance of transgenic lines [22]. Meanwhile, the research has found that Phosphate Starvation Response 1 (PHR1) interacts with the JASMONATE TIFY domain (JAZ) and MYC2, regulating jasmonate signaling induced by phosphate deficiency in *Arabidopsis thaliana* [23].

However, little is known about the response of *JAZ* family genes to abiotic stress in alfalfa. In this study, we identified *JAZ* gene family members in alfalfa by analyzing the evolutionary relationships, gene structure, protein motifs, and cis-acting elements of *MsJAZ* genes within the whole alfalfa genome. The expression patterns of the *MsJAZ* family members were analyzed in root, stem, leaf, and petiole tissues and under salt, drought, jasmonic acid (JA), and CrCl_3_•6H_2_O stresses. The results of this study lay a foundation for further understanding the functional verification of the *MsJAZ* gene family and provide a reference for exploring the mechanism of *JAZ* genes under abiotic stress in the future.

## 2. Results

### 2.1. Genome-Wide Identification of the MsJAZs Family of Alfalfa

To identify all JAZ family members in alfalfa (Zhongmu No. 1), protein sequences of 13 JAZ proteins in Arabidopsis were downloaded from TAIR as a query to search the alfalfa genome (https://figshare.com/articles/dataset/Medicago_sativa_genome_and_annotationfiles/12623960, accessed on 20 October 2024). After removing identical and incomplete gene sequences, a total of nine JAZ genes containing conserved structural domains of TIFY and Jas-motif (also known as CCT-2) were identified. These members were named *MsJAZ1* to *MsJAZ9* based on their location on the chromosome (Table 1). The nucleic acid and protein sequences of the members of the *MsJAZ* gene family are shown in Appendix A. MsJAZ protein sequences ranged in length from 94 aa (MsJAZ2) to 327 aa (MsJAZ7), and the corresponding protein molecular weights (MW) ranged from 19.33 to 38.03 kDa. The theoretical isoelectric point (pI) and instability index of MsJAZ ranged from 7.1 to 10.01 and 42.33 to 78.77, respectively, with a GRAVY of (−0.622) to (−0.15). The pIs of the MsJAZ proteins were all greater than 7 and are presumed to be basic proteins. The secondary structure analysis of MsJAZ proteins showed that the proportions of α-helices, extended strands, and random coils ranged from 9.49% (MsJAZ3) to 27.66% (MsJAZ2), 4.28% (MsJAZ1) to 12.77% (MsJAZ2), and 55.97% (MsJAZ2) to 89.53% (MsJAZ1). The predicted subcellular localization showed that most of the MsJAZ proteins were located in the nucleus, indicating that the JAZ family proteins mainly function in the nucleus. The 3D structures of the MsJAZ proteins are shown in Appendix A. Gene Ontology (GO) analysis revealed that all *MsJAZ* gene functions can be classified into biological processes and molecular functions, details of which are shown in Appendix A. The biological process category includes 10 terms, such as biological regulation, cellular process, developmental process, multicellular biological process, reproductive process, and response to stimuli. Molecular function categories include binding and transcriptional regulatory activity, with two and eight *MsJAZ* genes assigned to binding and transcriptional regulatory activity, respectively.

### 2.2. Phylogenetic Analysis of MsJAZ Proteins in Alfalfa

Studying the evolutionary relationships between species can help to understand the evolutionary relationships between species. To investigate the evolutionary relationships of JAZ proteins, a phylogenetic tree was constructed using JAZ family members from *M. sativa*, Arabidopsis, *Zea mays*, and *Oryza sativa*. As shown in Figure 1, a total of 56 JAZ proteins were classified into five clades (I, II, III, IV, and V) based on the similarity of the full-length sequences. Except for clade III, the other four clades contain JAZ proteins from all four species, indicating that JAZ proteins are highly conserved. Clade I consisted of 3 AtJAZs and 2 MsJAZs. Clade II consisted of 4 AtJAZs, 2 MsJAZs, 1 OsJAZ, and 2 ZmJAZs. Clade III had 5 ZmJAZs and 2 OsJAZs, but MsJAZ proteins were not identified in the same clade. Clade IV contained 3 AtJAZs, 2 MsJAZs, 2 OsJAZs, and 3 ZmJAZs. Clade V consisted of 3 AtJAZs, 3 MsJAZs, 6 OsJAZs, and 13 ZmJAZs. In addition, we found large differences in the number of genes in the *JAZ* gene family between the species and considerable interspecies variation in the *JAZ* gene family.

### 2.3. Functional Domains, Conserved Motifs, and Gene Structure Analysis of MsJAZ Genes

The *JAZ* family is a subfamily of the TIFY family of proteins that all contain the Jas-motif structural domain, which has the amino acid sequence SLX2FX2KRX2RX5PY and is involved in JA signaling. Further analysis of gene structure and conserved structural domains in *MsJAZ* is shown in Figure 2. Firstly, phylogenetic analysis of nine MsJAZs’ protein sequences was performed to classify MsJAZ proteins into three subfamilies (I, II, and III), distributed on chromosomes one, two, six, and eight (Figure 2E). All nine MsJAZ members of alfalfa contain conserved TIFY and Jas-motif functional domains (Figure 2B), with the TIFY structural domain preceding Jas-motif. Among them, MsJAZ7 contains two TIFY and Jas-motif functional domains. The 3D structures of the MsJAZ proteins are shown in Appendix A.

To better understand the conserved structure of the MsJAZ protein, the distribution of conserved motifs was identified and visualized using the MEME program and TBtools-II software, respectively. A total of 10 conserved motifs were identified in alfalfa MsJAZ proteins. The sequence information for each motif is provided in Appendix A. As shown in Figure 2C, a total of 10 conserved motifs were identified in alfalfa MsJAZ proteins. The results showed that all nine MsJAZs contained motif 1 and motif 2. In addition, MsJAZ5/6 and MsJAZ4/7 proteins had the same number and type of motifs, and they were close to each other on the phylogenetic tree, indicating a high degree of homology (Figure 2A).

The structure of a gene determines the function of a protein, and therefore, gene structure plays an important role in the regulation of gene expression. By analyzing the structure of *MsJAZ* genes, we found that all members of the *MsJAZ* gene family have introns and exons, but the number of them varies greatly (Figure 2D). For example, *MsJAZ6* contains seven introns and eight exons, whereas *MsJAZ7* contains only one intron and two exons. In addition, *MsJAZ1/2/3/5* all contain two UTRs, *MsJAZ4* and *MsJAZ7* contain one UTR, and *MsJAZ6/8/9* do not have a UTR structure.

### 2.4. Analysis of Cis-Acting Elements in the MsJAZ Gene Promoters

Analysis of promoter *cis*-elements can provide ideas for tissue-specific expression of genes and stress response patterns. We used Plant Care to analyze *cis*-acting elements in the promoters of *MsJAZ* members. Figure 3 displays the results, which showed that the *MsJAZ* genes had a total of 27 different types of *cis*-acting elements grouped into three groups (Appendix A). These elements were found to play a major role in growth, phytohormone response, light response, and stress response. Light-responsive elements included the main ABRE, CGTCA-motif, AAGAA-motif, as-1, MYC, ERE, TGA-element, TGACG-motif, and TCA-element; phytohormone-responsive elements included the ABRE, CGTCA-motif, AAGAA-motif, as-1, MYC, ERE, TGA-element, TGACG-motif, and TCA-element; and stress-responsive elements comprised G-box, W box, ARE, MBS, WUN-motif, DRE core, LTR, STRE, and WRE3.

### 2.5. Analysis of the Chromosomal Localization and Collinearity of MsJAZ Genes

Alfalfa genome annotation determined that *MsJAZ* genes are distributed only on chromosomes 1, 2, 6, and 8. As shown in Figure 4A, the nine *MsIAZ* genes were unevenly distributed on chromosomes 1, 2, 6, and 8 of alfalfa. Chromosomes 1 and 6 each contained one *MsJAZ* gene, chromosomes 2 contained four *MsJAZ* genes, and chromosomes 8 contained three *MsJAZ* genes. Meanwhile, to further investigate the evolutionary mechanism and homology of the MsJAZ gene family, we analyzed tandem and segmental duplication events. We found that *MsJAZ4/MsJAZ7*, *MsJAZ5/MsJAZ6*, *MsJAZ7/MsJAZ9*, *MsJAZ5/MsJAZ8*, and *MsJAZ4/MsJAZ9* were segmental duplications (Table 2). In addition, we analyzed the covariance of the *JAZ* genes in alfalfa, Arabidopsis, maize, and *Medicago truncatula*. Notably, the covariance was closer between alfalfa and *Medicago truncatula* than between alfalfa and Arabidopsis or maize, suggesting that alfalfa and *Medicago truncatula* have a high degree of homology.

### 2.6. Expression Patterns of MsJAZ Genes in Different Tissues

To further investigate the expression patterns of *MsJAZ* genes in alfalfa, RT-qPCR analysis was performed to examine the expression levels of these genes in different tissues, including roots, stems, leaves, and petioles (Figure 5A). The results revealed that the expression profiles of *MsJAZ* genes could be categorized into three distinct types based on their expression patterns (Figure 5A). The first type, which includes *MsJAZ1/3/7/9*, exhibited the highest expression levels in stems. The second type, comprising *MsJAZ1/2/7/8/9*, showed predominant expression in leaves (Figure 5A). The third type, represented by *MsJAZ1/4/7/8/9*, displayed the highest expression levels in young stems and relatively high expression in mature leaves (Figure 5A). In summary, the majority of *MsJAZ1/4/9* genes were predominantly expressed in stems, leaves, and petioles, suggesting their potential role in regulating the growth and development of alfalfa (Figure 5). Additionally, correlation analysis revealed that the expression patterns of most *MsJAZ* genes were positively correlated. However, only a few *MsJAZ* genes exhibited a significant direct correlation (Figure 5B).

### 2.7. Expression Patterns of MsJAZ Genes in Response to Abiotic Stress

To explore the potential roles of *MsJAZ* genes in response to various abiotic stresses, their expression profiles in alfalfa leaves under salt and drought treatment were analyzed using RT-qPCR. The results show that the relative expression levels of all *MsJAZs* were significantly unregulated, although the response time and the extent of up-regulation varied (Figure 6). Under salt treatment, the expression levels of *MsJAZ4/7/9* peaked at 1 h; the expression levels of *MsJAZ1/3/5/6* reached their maximum at 3 h, with fold changes of 2.18, 4.24, 4.84, and 3.74 compared to 0 h, respectively, and the expression levels of *MsJAZ2* and *MsJAZ8* peaked at 6 h (Figure 6A). Under drought treatment, all *MsJAZs* except *MsJAZ8* reached their maximum expression at 12 h. *MsJAZ4, MsJAZ5*, and *MsJAZ7* showed a more pronounced response to drought, with expression levels 30.72, 3.97, and 10.97 times higher than those at 0 h, respectively (Figure 6B). Overall, *MsJAZ4* and *MsJAZ7* exhibit a highly pronounced response to both salt and drought treatment.

### 2.8. Expression Profile of MsJAZ Genes in Response to Chromium and MeJA

The proliferation of industrial activity has resulted in an escalation in chromium (Cr) contamination of the environment. It is imperative to acknowledge the significance of chromium (Cr) pollution as a significant source of heavy metal stress, which must be addressed as a primary concern among the various abiotic stresses that have been identified. To investigate the potential role of *MsJAZ* genes in response to heavy metal and hormone stress, we employed RT-qPCR to analyze their expression profiles in alfalfa leaves under chromium and jasmonic acid treatment. The results show that the relative expression levels of most *MsJAZ* genes were significantly up-regulated, but the response times and extents of up-regulation varied (Figure 7). Under chromium treatment, the expression levels of *MsJAZ3/4/5/6/7/9* peaked at 12 h (Figure 7A). Under jasmonic acid treatment, the expression levels of *MsJAZ5* and *MsJAZ8* increased with the duration of treatment, while *MsJAZ3/4/5/6/7/8/9* reached their maximum expression at 12 h (Figure 7B). *MsJAZ4* expression peaked at 3 h of chromium and jasmonic acid treatment, while *MsJAZ7* and *MsJAZ9* peaked at 12 h. Overall, the responses of *MsJAZ4* and *MsJAZ9* to chromium and jasmonic acid treatment were relatively pronounced.

### 2.9. MsJAZ Protein Interaction Network Prediction and Subcellular Localization

To provide further evidence for the function of the MsJAZ protein, the online STRING database was utilized to predict its function. The results of this analysis indicated that MsJAZ2, MsJAZ4, MsJAZ8, and MsJAZ9 interacted with each other, except for MsJAZ3, which did not interact with MsJAZ9 (Appendix A). Furthermore, the MsJAZ proteins were found to interact with the repressors of the jasmonate response (TIFY and JAZ), the negative regulator of the jasmonate response (AFPH2), the transcription factor bHLH13, and the GATA transcription factor 25 (Appendix A).

Studies of the subcellular localization of proteins are useful for the elucidation of their functions. Members of the JAZ family, such as the phytohormone jasmonates, are characterized by the presence of the TIFY and CCT_2 structural domains and are localized in nuclear structures. To further validate the accuracy of the subcellular localization predictions, the researchers selected the *MsJAZ4* and *MsJAZ7* genes, which are highly responsive to abiotic stresses, for the detection of transient expression in tobacco (Figure 8). In tobacco leaves injected with the control vector pCAMBIA1300-GFP, green fluorescence was observed throughout the cells. However, in tobacco leaves injected with the pCAMBIA1300-MsJA4/7-GFP fusion vector, fluorescence was observed at both the plasma membrane and the nucleus. In addition, co-localization with the plasma membrane and NLS-mCherry markers showed that green and red fluorescence could merge. This suggests that MsJAZ4 and MsJAZ7 proteins were localized to the nucleus and cell membrane.

### 2.10. Functional Validation of the MsJAZ Gene in Yeast

To identify the function of the *MsJAZ* gene, the CDS sequence of the *MsJAZ4/7* gene was cloned using pYES2-MsJAZ4/7-F/R as a primer and then ligated into the pYES2 vector by the homologous arm recombination method using the ClonExpress MultiS One Step Cloning Kit from Novozymes Biologics. It was then ligated into the pYES2 vector by the homology arm recombination method. The correctly verified positive clones were incubated in liquid medium SD/-Ura containing 2% galactose with shaking and then coated on solid SG/-Ura medium containing 200 mM mannitol and 200 mM NaCl in a gradient of 10, 10^−1^, 10^−2^ and 10^−3^, respectively. The results showed that *MsJAZ4* and *MsJAZ7* genes were more sensitive to drought and salt stress (Figure 9).

## 3. Discussion

Jasmonic-L-isoleucine (JA-Ile) is an important signal molecule. JAZ proteins have been identified as inhibitors of the JA pathway, which regulates relevant physiological activities including anthocyanin accumulation, plant defense responses [24], flowering time regulation [25], stamen development, and cold stress responses by inhibiting JA-responsive TF expression [26]. Due to the continuous development of biotechnology, JAZ proteins have been identified in Arabidopsis, tomato [27], maize [28], soybean [29], wheat [30], rice [31], and Camellia species [32]. However, to date, the JAZ proteins in alfalfa, in particular, have been the subject of less research. Therefore, in this study, the *JAZ* gene family of alfalfa (Zhong clover I) was comprehensively identified, and the gene structure, functional properties, promoter *cis*-elements, expression pattern, abiotic stress, and subcellular location were thoroughly investigated to lay a foundation for gene function studies of *MsJAZs*.

Structural domains are regions of a protein molecule with a specific structure and function and are the basic functional units of proteins [33]. Previous studies have shown that the JAZ proteins have conserved structural domains in both the TIFY and CCT_2 (also known as Jas) domains. A total of nine *JAZ* genes have been identified in the genome of alfalfa (Zhong clover I). These genes have been named *MsJAZ1*-*MsJAZ9*. The number of *JAZ* genes in alfalfa was lower than in Arabidopsis (n = 13) [34] and *Camellia sinensis* (n = 13) [35], rice (n = 15) [36], maize (n = 16) [28], bread wheat (n = 14) [37], and soybean (n = 33), and there were the same number of tomato (n = 9) [12] *JAZ* genes. MsJAZ proteins differed in length, molecular weight (MW), and theoretical isoelectric point (PI). Through subcellular localization prediction, we found that most of the MsJAZ proteins were located in the nucleus. Based on the results of the GO analysis, all *MsJAZ* genes were found to be involved in various biological processes and molecular functions, suggesting their significant role in the growth and development of alfalfa.

Phylogenetic analysis indicates that the nine MsJAZ proteins can be divided into four subfamilies based on their evolutionary relationships with Arabidopsis JAZ proteins. The prominent clustering of specific JAZ proteins from alfalfa and Arabidopsis within the same branch suggests that alfalfa is genetically more closely related to Arabidopsis than to maize or rice. Genes with high homology were likely to share similar functions, allowing predictions of gene functions based on homology with other species and phylogenetic analyses. *AtJAZ1* and *OsJAZ9* have been identified as regulators of salt stress, suggesting that MsJAZ proteins, which exhibit high homology with *AtJAZ1* and *OsJAZ9* in the phylogenetic tree, may also play a role in salt stress regulation. All *MsJAZ* genes contain both TIFY and JAS domains, with their positions aligning with previous studies based on NCBI-CDD results. Most of the MsJAZ proteins share similar 3D structures, suggesting that they may have similar functions. The structure of exons and introns affects protein function, gene expression, and regulation. Gene structure analysis revealed that *MsJAZ* genes share similar motifs and intron/exon arrangements, suggesting that genes within the same subfamily may have subfamily-specific functions.

In this study, the *MsJAZ* genes were found to be dispersed across diverse chromosomes of alfalfa. Notably, their distribution pattern was non-uniform, a characteristic shared with the majority of species [38]. Gene duplication is the primary mechanism for the generation of new genes and ultimately new biological functions [39] and is critical for plant evolution and adaptation, standing as one of the pivotal forces propelling genome evolution and species genetics [40]. It not only augments gene families by incorporating new members but also enriches their functional repertoires, significantly fueling the genetic advancement of assorted organisms. In plants, the expansion of gene families predominantly stems from two main sources: segmental duplication and tandem duplication [41]. Within the *MsJAZ* genes family, the occurrence of segmental duplication events was noted, signifying that gene duplication acts as a catalyst for the proliferation of *MsJAZ* gene family constituents as well. On the other hand, both Ka/Ks of the *MsJAZ* genes were less than one, suggesting that the *MsJAZ* genes may have been under purifying selection during their evolution. Simultaneously, this reflects the central position and functional constraints of the *MsJAZ* genes in the jasmonate pathway. This evolutionary conservatism ensures that plants maintain a stable defense response in complex environments, while at the same time balancing the need for conservatism and adaptability through the expansion of gene families (multicopy) and flexibility in the regulation of expression. Prior investigations have revealed that gene pairs with pronounced homology within a gene family tend to display collinearity, thereby conferring upon them analogous biological traits and expression modalities. In the current study, it was ascertained that MsJAZ exhibited collinearity with both alfalfa and Arabidopsis, hinting at the potential for shared functions, albeit necessitating further in-depth exploration. Intriguingly, a greater number of collinear gene pairs were detected between alfalfa samples compared to those between Arabidopsis, suggesting a closer genetic relationship and laying a solid groundwork for predicting the expression and function of *MsJAZ* genes (Figure 4B).

The study of gene tissue specificity is a critical step in understanding life processes and tissue functions [42]. In this research, we observed that all *MsJAZ* genes in alfalfa exhibited tissue-specific expression patterns (Figure 5A), similar to findings in Mentha canadensis [43] and Petunia progenitors [44]. Most *MsJAZ* genes were expressed in both leaves and petioles, suggesting their potential roles in vegetative growth. Notably, *MsJAZ1*, *MsJAZ7*, and *MsJAZ9* were expressed in stems, leaves, and petioles (Figure 5A), while *MsJAZ3* showed no detectable expression in these tissues (Figure 5A). Interestingly, *MsJAZ2* was exclusively expressed in leaves. In summary, the predominant expression of most *MsJAZ* genes in leaves and petioles led us to hypothesize that these genes may be involved in the reproductive growth and development of alfalfa.

Drought and salt stress are among the most prevalent abiotic challenges faced by plants [45]. To cope with these adversities, plants have developed adaptive mechanisms that involve the induction of stress-related gene expression. These genes play a critical role in modulating physiological responses, thereby enhancing the plant’s tolerance to such stresses [46,47]. The *JAZ* genes have been found to play a role in regulating plant tolerance in response to both abiotic and biotic stresses. In this study, nine *MsJAZ* members exhibited a positive response to salt stress (Figure 6A), indicating that the *MsJAZ* genes may play a regulatory role in osmotic stress. Except for *MsJAZ2*, the expression levels of all *MsJAZ* genes were significantly up regulated under drought stress (Figure 6B), suggesting a positive response to drought conditions. Our detailed analysis revealed that *MsJAZ4*, *MsJAZ7*, and *MsJAZ9* were significantly up regulated under both drought and salt stress (Figure 6), consistent with the predicted results of promoter regulatory elements. Notably, *MsJAZ4* and *MsJAZ7* displayed similar expression patterns under salt and drought stress, likely due to their close phylogenetic relationship in the evolutionary tree. This study found that overexpression of *GsJAZ2* [22] and *MdJAZ2* [48] in *Arabidopsis thaliana* significantly enhanced tolerance to salt and alkali stress. Similarly, *OsJAZ1* plays a role in regulating drought resistance in rice, partly through the ABA and JA pathways. For example, rice plants overexpressing *OsJAZ1* exhibited greater sensitivity to drought stress during both the seedling and reproductive stages compared to the wild-type Zhonghua 11 (ZH11). In contrast, jaz1 T-DNA insertion mutant plants showed higher drought tolerance than WT plants [21]. Furthermore, *PnJAZ1* has been shown to promote plant growth under salt stress by mediating the abscisic acid signaling pathway [49]. Based on these findings, we hypothesize that the *MsJAZ* genes may be involved in regulating osmotic stress induced by drought and salt conditions.

Chromium (Cr) is a naturally occurring heavy metal [50]. Although plants and animals require trace amounts of Cr, at higher concentrations it remains a major environmental pollutant. However, with industrialization, pollution by the heavy metal chromium is becoming a serious environmental problem throughout the world. Delayed seed germination, inhibition of plant growth, photosynthetic damage, chlorosis and necrosis of leaves, low seed yield, and finally plant death are the effects of the heavy metal Cr on plants [51]. Heavy metals affect endogenous levels of jasmonates, and jasmonates regulate different plants under heavy metal stress [52]. In this study, the expression levels of *MsJAZ4/7/9* were significantly up regulated under Cd stress. The results of this study support the hypothesis that heavy metal conditions activate *JAZ* gene expression. It was found that endogenous JA accumulation increased in pepper plants under Cd stress [53] and that JA improved the effects of Cr-mediated phytotoxicity on gas exchange and photosynthetic pigments on plant growth [54]. The expression patterns of *MsJAZ4/7* and *MsJAZ5/6* in jasmonic acid treatment exhibited a high degree of similarity, suggesting a potential correlation with their position in the same branch of the evolutionary tree (Figure 7B). Jasmonates (JAs) play a crucial role in plant responses to both biotic and abiotic stresses, serving as a key regulator in plant defense mechanisms. As a signaling molecule, JA orchestrates the plant’s reactions to various stresses by inducing the expression of resistance genes and enhancing stress tolerance. In this study, under jasmonate treatment, the expression level of *MsJAZ* genes was significantly increased, indicating a positive response of *JAZ* genes to jasmonate.

## 4. Materials and Methods

### 4.1. JAZ Gene Identification in Alfalfa

The protein sequences of 13 Arabidopsis JAZ genes were downloaded from TAIR. The reference genome in alfalfa (Zhongmu No. 1) and annotation files (https://figshare.com/articles/dataset/Medicago_sativa_genome_and_annotation_files/12623960, accessed on 20 October 2024) were used in this study. Firstly, 13 AtJAZ protein sequences were used to perform a BLAST search against the alfalfa protein sequences used TBtools-II software, with the threshold of E-value < 1 × 10^−5^. Furthermore, the Hidden Markov Model (HMM) profiles of PF06200 and PF09425 were used as a query to search against the alfalfa protein database using the simple HMM search in TBtools [55]. Subsequently, the BLAST and HMMER search results were merged, and redundancies were removed manually. Then, the InterPro (https://www.ebi.ac.uk/interpro/search/sequence/, accessed on 22 October 2024) function was used to confirm whether the candidate *MsJAZs* had the conserved TIFY and Jas-motif. We deleted the gene without the typical functional domain.

### 4.2. Basic Analysis of MsJAZ Proteins

We analyzed the protein length (aa), instability index, molecular weight (MW), isoelectric point (pI), and grand average of hydropathicity (GRAVY) of MsJAZ proteins using the ProtParam ExPASy online website (https://web.expasy.org/protparam/, accessed on 25 October 2024). The SOPMA (https://npsa-prabi.ibcp.fr/cgi-bin/secpred_sopma.pl, accessed on 28 October 2024) was used to analyze the secondary structure of MsJAZ proteins. The Plant-mPLo (http://www.csbio.sjtu.edu.cn/bioinf/plant-multi/, accessed on 5 November 2024) was used to predict MsJAZ proteins subcellular localization [56]. We used the AlphaFold database to analyze the three-dimensional (3D) structure of MsJAZ proteins [57].

### 4.3. Phylogenetic Analysis

The phylogenetic tree of 56 JAZ protein sequences, including 9 MsJAZs from alfalfa, 13 AtJAZs from Arabidopsis, 12 OsJAZs from *Oryza sativa*, and 22 ZmJAZs from *Zea mays*, was constructed using MEGA 11 software with maximum likelihood estimate (ML), bootstrap values of 1000 times, and other parameters set by default [58].

### 4.4. Functional Domain, Conserved Motifs and Gene Structure Analysis

The InterPro (https://www.ebi.ac.uk/interpro/, accessed on 7 November 2024) online website was used to analyze the functional domains of MsJAZ proteins. We used the MEME (https://meme-suite.org/meme/tools/meme, accessed on 10 November 2024) to analyze conserved motifs and set the maximum number of predicted patterns to 10, and the screening threshold was E < 1 × 10^−10^ [59]. The gene structure was analyzed using the alfalfa genome annotation files and visualized with TBtools.

### 4.5. Promoter Cis-Acting Element Prediction

The 2 kb genomic DNA sequence upstream of the start codon (ATG) of each MsJAZ gene was extracted from the alfalfa genome annotation files using TBtools. The *cis*-regulatory elements in the promoter sequences of *MsJAZ* genes were analyzed by PlantCare (http://bioinformatics.psb.ugent.be/webtools/plantcare/html/, accessed on 12 November 2024) [60]. Arabidopsis was used as a model plant for the query, and the STRING online website (http://cn.string-db.org/, accessed on 15 November 2024) was used to analyze the protein–protein interactions of MsJAZs.

### 4.6. Chromosome Location, Gene Duplication, and Collinearity Analysis

The genomic data for Arabidopsis and maize were downloaded from the EnsemblPlants database (https://plants.ensembl.org/index.html, accessed on 15 November 2024). Chromosome distribution of *MsJAZ* genes was analyzed and localized using TBtools-II software (https://github.com/CJ-Chen/TBtools, accessed on 17 November 2024). The multiple collinear scanning toolkit (MCScanX) was utilized to analyze the collinear blocks of *JAZ* genes across alfalfa, Arabidopsis, *Zea mays*, and *Medicago truncatula* and visualized by TBtools. Synonymous (Ka) and non-synonymous (Ks) substitutions and Ka/Ks ratios were calculated by the Simple Ka/Ks Calculator (NG) [61].

### 4.7. Plant Materials and Stress Treatments

Alfalfa (Zhongmu No. 1) seeds were sterilized with 75% ethanol for 30 sec and 50% sodium hypochlorite for 5–6 min, then rinsed 10 times with distilled water and incubated on wet filter paper in Petri dishes at 25 °C. After 3 days of growth in Petri dishes, alfalfa seedlings were transferred to a plastic container (25 cm × 18 cm × 14 cm) with Hoagland nutrient solution for hydroponics under controlled conditions: temperature 24 ± 1 °C, 16 h of light, 8 h of darkness, and 60% relative humidity. After 30 days of growth, the seedlings were exposed to nutrient solutions supplemented with 20% PEG-6000, 200 mM NaCl, 100 μM JA [62], or 300 μM CrCl_3_•6H_2_O for different treatments. The seedlings cultured with a normal nutrient solution without adding any substance served as the control. The leaf samples from the control and treatment groups of alfalfa were collected at five time points of 0, 1, 3, 6, and 12 h for gene expression analysis. The roots, stems, leaves, and petioles of the untreated seedlings were collected for gene expression analysis in different tissues. All samples were rapidly frozen in liquid nitrogen and stored at −80 °C for further RNA extraction.

### 4.8. RNA Extraction and Quantitative Real Time PCR (RT-qPCR) Analysis

Total RNA was extracted from the samples using the RNA Extraction Kit (Promega, Shanghai, China) according to the instructions. A total of 1 μg RNA was used for reverse transcription, and RT-qPCR was carried out according to the method provided by ChamQ Universal SYBR qPCR Master Mix (Vazyme, Nanjing, China). The MsActin gene [63] was used as an internal reference gene, and all primers used in this study are listed in Appendix A. All primer concentrations were 10 μg/mL. The RT-qPCR reaction system was 20 μL, containing 2 μL of cDNA, 0.6 μL of forward and reverse primers, 10 μL of ChamQ Universal SYBR enzyme, and 6.8 μL of ddH_2_O (no RNA enzyme water). The reaction procedure was 3 min at 95 °C; 10 s at 95 °C, 30 s at 60 °C, and 30 s at 72 °C for 40 cycles; 15 s at 95 °C, 1 min at 60 °C, and 5 s at 95 °C. Each treatment for RT-qPCR consisted of three independent biological replicates and three technical replicates. We normalized the expression values of the root tissue at 0 h and evaluated the relative expression values of other tissues at different time points. The relative expression of genes was calculated by 2^−ΔΔCT^ [64].

### 4.9. Subcellular Localization of MsJAZ Proteins

Based on the results of the *MsJAZ* genes’ response to abiotic stress, *MsJAZ* was selected for amplification, and the primers are shown in Appendix A. After PCR product recovery and purification, the CDS clones were inserted into the BamH1 and Sac1 cloning sites in the pCAMBIA-1300-GFP vector (NovoPro, Shanghai, China). The recombinant vector was separately transformed into Agrobacterium tumefaciens GV3101. To validate the transient expression of *MsJAZ* in tobacco, Agrobacterium with mCherry and pm-rk strains carrying the recombinant vectors were injected into tobacco leaves grown for one month. After incubating in the dark for 48 h, the leaf epidermal cells were used for microscopic (Leica SP8, Wetzlar, Germany) observation and image acquisition by the laser confocal method.

### 4.10. Functional Validation of the MsJAZ Protein

The CDS clone of MsJAZ4/7 was inserted into the BamH1 and EcoR1 cloning sites of the PYES2 vector to verify the potential function of the *MsJAZ4/7* gene. Following the transformation of the recombinant vectors into Saccharomyces cerevisiae INVSC1, the positive monoclines were shaken in 2% glucose liquid medium (SD-U) until OD 1.2–1.4, centrifuged, and the organisms were collected. After adjusting OD to 1.0 in the subsequent 1:1, 1:10, 1:100, and 1:1000 cycles, the organisms were then incubated in 2% galactose liquid medium SD-U for 8–12 h. After adjusting the OD to 1.0, plates were spotted onto a solid medium containing 200 mM NaCl and 200 mM mannitol at 10, 10^−1^, 10^−2^, and 10^−3^ gradients; strain growth was observed, and photographs were taken to record results.

### 4.11. Statistical Analysis

Origin 2024 software was used to test the differences between groups through one-way analysis of variance (ANOVA). Tukey’s multiple range tests were used to assess the significant differences; *p* < 0.05 indicates significance. Data were expressed as mean or mean ± standard deviation (SD).

## 5. Conclusions

In conclusion, a total of nine JAZ family proteins located on four different chromosomes have been identified from the alfalfa (Zhongmu No. 1) genome. Phylogenetic analyses showed that these nine proteins could be classified into five groups (clade I–V), which were supported by their exon/intron structures, motifs, and structural domains. All MsJAZ proteins have nuclear TIFY and Jas-motif structural domains. RT-qPCR results showed tissue-specific expression of all MsJAZs, with MsJAZ4 and MsJAZ7 showing the strongest response to salt and drought. Meanwhile, MsJAZ4, MsJAZ7, and MsJAZ9 responded positively to chromium and MeJA, suggesting that they may have functions. This study provides a basis for further research into the functions and regulatory mechanisms of MsJAZs.

## Figures and Tables

**Figure 1 ijms-26-04684-f001:**
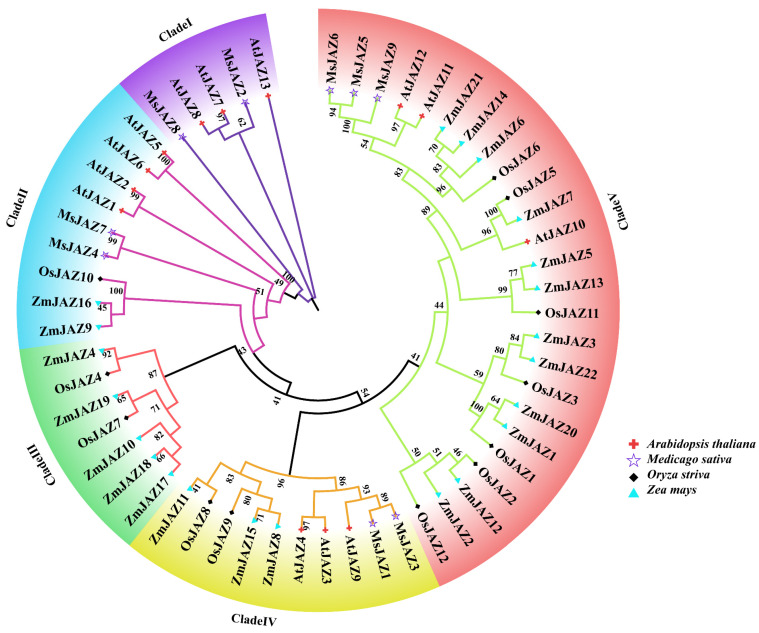
Phylogenetic tree analysis of JAZ proteins in Arabidopsis, *M. sativa*, *Oryza sativa*, and *Zea mays*. The phylogenetic tree was constructed using JAZ protein sequences by the maximum likelihood estimate (ML) in MEGA 11 with 1000 bootstrap replicates. The red crosses, purple stars, black diamonds, and blue triangles represent the Arabidopsis, *M. sativa*, *Oryza sativa*, and *Zea mays* JAZ proteins, respectively.

**Figure 2 ijms-26-04684-f002:**
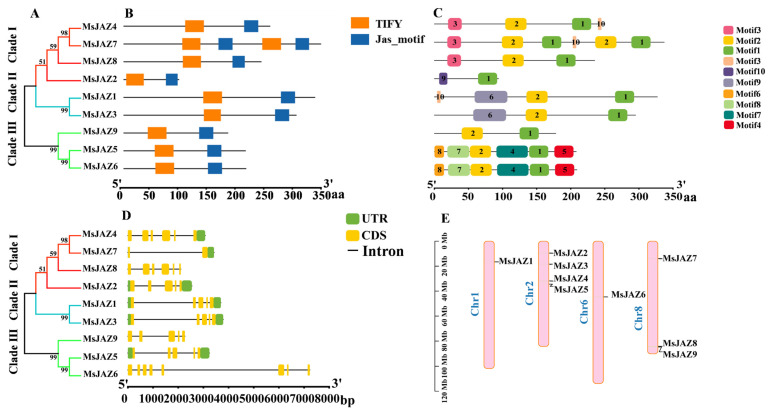
Analysis of the conserved domains, motifs, and gene structure of *MsJAZs*. (**A**) Phylogenetic tree of the *MsJAZ* gene family. (**B**) Functional domains distribution of MsJAZs. The colored rectangles represent the protein-conserved domains. (**C**) Motifs of MsJAZ proteins. Different motifs are annotated by boxes of different colors and numbered 1–10. (**D**) The exon–intron structure of *MsJAZ* genes. (**E**) Chromosome of *MsJAZ* genes. The untranslated regions (UTRs) are indicated by green boxes. Yellow boxes represent exons, and black lines represent introns.

**Figure 3 ijms-26-04684-f003:**
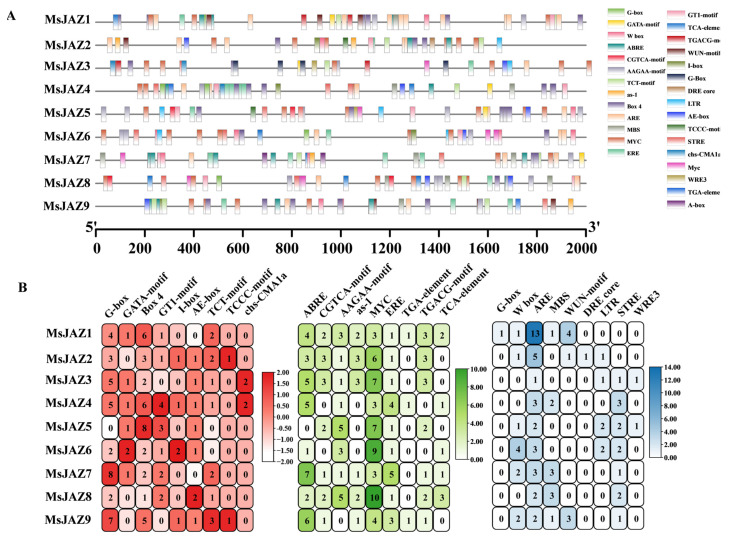
Distribution of *cis*-acting elements in the promoter region of *MsJAZ* genes in alfalfa. (**A**) TBtools was utilized for the visualization of homeotic elements in the promoter of *MsJAZs*, encompassing their position, type, and number. (**B**) The present study will examine the number of different components in the light signaling response, phytohormone response, and stress-related components.

**Figure 4 ijms-26-04684-f004:**
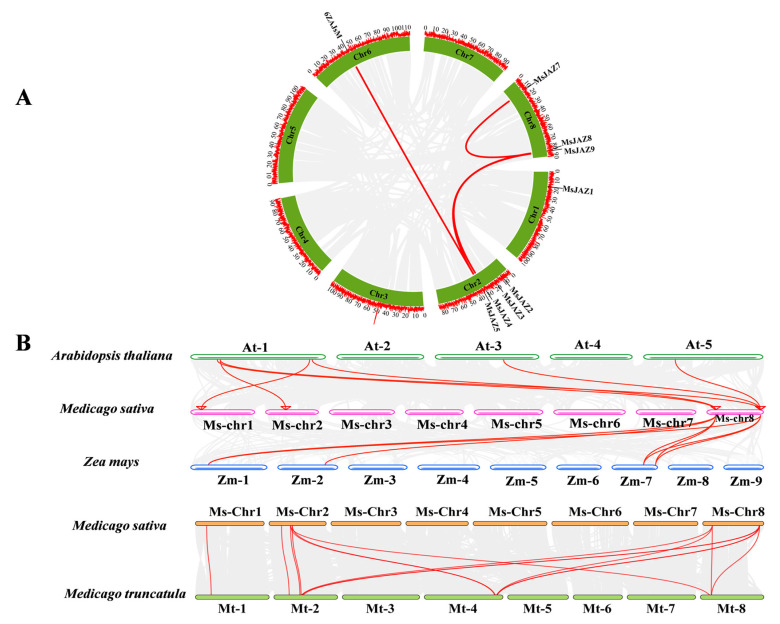
Chromosomal distribution and collinearity analysis of MsJAZ genes. (**A**) Chromosomal localization and intrachromosomal relationships in alfalfa. The black lines represent the MsJAZ collinearity genes. (**B**) Collinearity analysis of *JAZ* genes in alfalfa, Arabidopsis, *Zea mays*, and *Medicago truncatula*; the grey lines in the background represent collinear blocks between alfalfa and Arabidopsis/*Zea mays*/*Medicago truncatula*, and the red lines represent collinear JAZ gene pairs.

**Figure 5 ijms-26-04684-f005:**
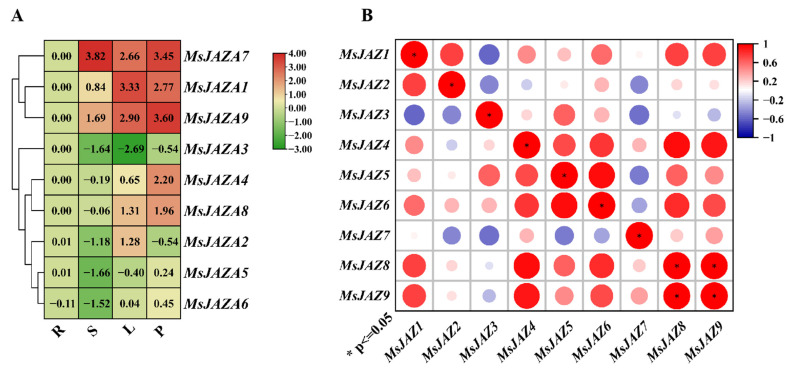
Expression analyses of *MsJAZ* genes across six alfalfa tissues. (**A**) A heat map illustrating the differential expression of *MsJAZ* genes in four tissues. R: roots, S: stems, L: leaf, P: petiole. The values represent log-transformed expression levels, with green and red indicating low and high expression, respectively. (**B**) A correlation heat map depicting expression patterns across the four tissues. Correlations are represented by blue and red colors, signifying negative and positive correlations, respectively. Significance was analyzed by analysis of variance (ANOVA). The values represent log_2_ fold change expression levels, with green and red indicating low and high expression, respectively.

**Figure 6 ijms-26-04684-f006:**
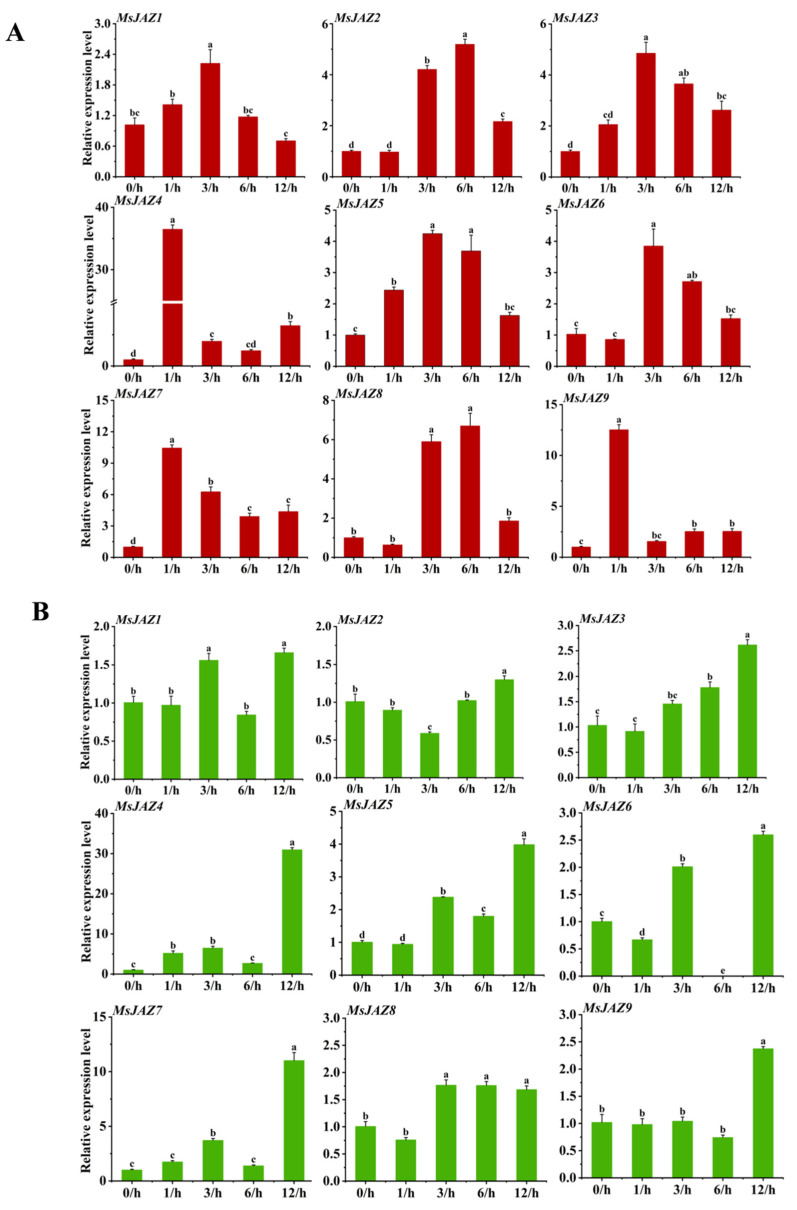
The relative expression levels of *MsJAZ* genes under abiotic stress at 0 (CK), 1, 3, 6, and 12 h. (**A**) Relative expression levels of *MsJAZ* genes after salt treatment. (**B**) Relative expression levels of *MsJAZ* genes after drought treatment. The data are presented as the mean ± SE of three independent biological replicates and three technical replicates. The horizontal line above the columns represents the standard deviation. Letters indicate significant differences (*p* < 0.05).

**Figure 7 ijms-26-04684-f007:**
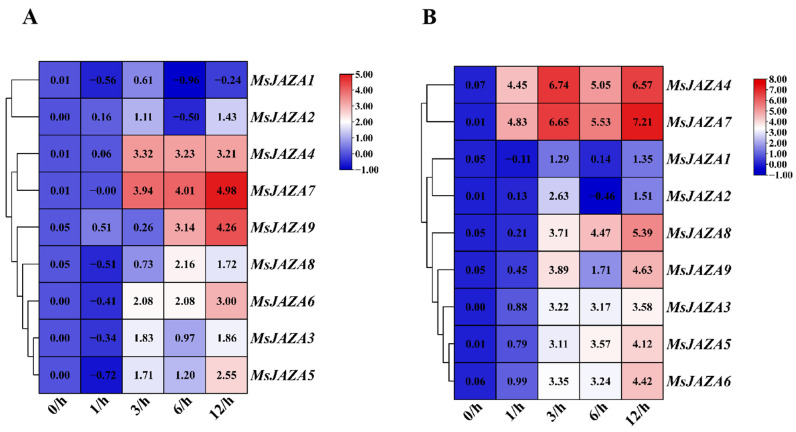
The relative expression levels of *MsJAZ* genes under abiotic stress at 0 (CK), 1, 3, 6, and 12 h. (**A**) Relative expression levels of *MsJAZ* genes after chromium treatment. (**B**) Relative expression levels of *MsJAZ* genes after jasmonic acid treatment. Red indicates an upward adjustment; black represents a downward adjustment.

**Figure 8 ijms-26-04684-f008:**
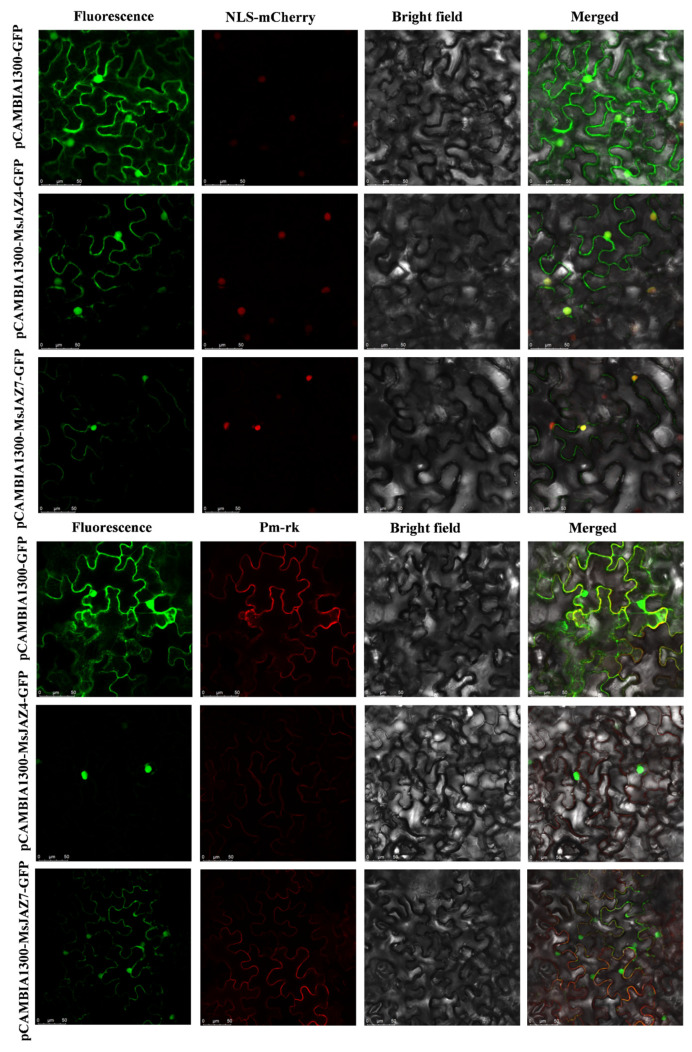
The subcellular localization of the MsJAZ4/7 protein. pCAMBIA1300-MsJAZ4/7-GFP protein fusions transiently expressed in tobacco leaf cells. Excitation light wavelength: GFP field: 488 nm, Pm-rk field: 587 nm, NLS-m Cherry: 587 nm. Scale bar: 50 μm.

**Figure 9 ijms-26-04684-f009:**
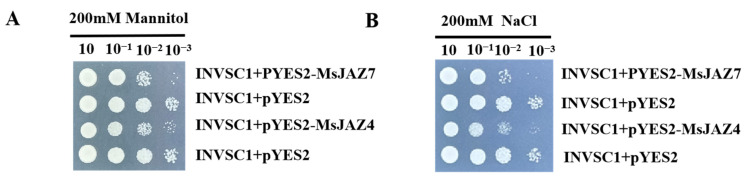
Growth images of pYES2 and pYES2-MsJAZ4/7 yeast strains on SG/-Ura solid medium containing 200 mM mannitol and 200 mM NaCl. (**A**) 200 mM mannitol; (**B**) 200 mM NaCl.

**Table 1 ijms-26-04684-t001:** Information on MsJAZ proteins in alfalfa.

Gene Name	Gene ID	Chr	Protein Length (aa)	Instability Index	MW (kDa)	pI	GRAVY	α- Helix (%)	Extended Strand (%)	Random Coil (%)	Subcellular Localization
											Cell-PLoc
*MsJAZ1*	MsG0180001147.01.T01	1	327	45.46	34.29	9.45	−0.15	9.79	4.28	85.93	Cell membrane. Nucleus.
*MsJAZ2*	MsG0280007019.01.T05	2	94	72.83	10.76	10.01	−0.622	27.66	12.77	59.57	Nucleus.
*MsJAZ3*	MsG0280007633.01.T01	2	294	48.97	31.03	7.1	−0.217	9.49	4.75	85.76	Cell membrane. Nucleus.
*MsJAZ4*	MsG0280008436.01.T01	2	250	44.96	27.15	8.83	−0.443	12.4	7.2	80.4	Nucleus.
*MsJAZ5*	MsG0280008590.01.T01	2	208	67.64	22.36	8.41	−0.42	9.62	5.29	85.1	Nucleus.
*MsJAZ6*	MsG0680032555.01.T01	6	209	78.77	22.41	8.52	−0.409	11	6.7	82.3	Nucleus.
*MsJAZ7*	MsG0880042776.01.T01	8	337	44.33	38.03	8.34	−0.436	12.46	5.04	82.49	Nucleus.
*MsJAZ8*	MsG0880047288.01.T01	8	235	48.71	25.91	8.34	−0.722	16.17	5.11	78.72	Nucleus.
*MsJAZ9*	MsG0880047330.01.T01	8	178	58.06	19.33	8.97	−0.433	9.55	5.06	85.39	Nucleus.

Note: aa—number of amino acids sequence; MW—theoretical molecular weight of proteins; pI—theoretical isoelectric point of proteins; GRAVY—grand average of hydropathicity.

**Table 2 ijms-26-04684-t002:** The divergence between *MsJAZ* gene pairs in alfalfa.

Paralogous Pairs	Ka	Ks	Ka/Ks	Duplicate Date (Mya)	Duplicate Type
*MsJAZ4/MsJAZ7*	0.64	1.38	0.46	113.49	segmental
*MsJAZ5/MsJAZ6*	0.02	0.04	0.47	3.15	segmental
*MsJAZ7/MsJAZ9*	0.61	2.20	0.28	180.37	segmental
*MsJAZ5/MsJAZ8*	0.39	0.58	0.68	47.87	segmental
*MsJAZ4/MsJAZ9*	0.32	0.80	0.40	65.24	segmental

Note: For each gene pair, the Ks value was translated into divergence time in millions of years based on a rate of 6.1 × 10^−9^ substitutions per site per year. The divergence time (T) was calculated as T = Ks/(2 × 6.1 × 10^−9^) × 10^−6^ Mya.

## Data Availability

All of the datasets supporting the results of this article are included within the article and Appendix A.

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
