# Peer review of "Genome-Wide Analysis and Expression Profiling of the JAZ Gene Family in Response to Abiotic Stress in Alfalfa"

_ijms, 2025, doi:10.3390/ijms26104684_

Round 1

Reviewer 1 Report

Comments and Suggestions for Authors

The authors conducted a study titled "Genome-Wide Analysis and Expression Profiling of the JAZ Gene Family in Response to Abiotic Stress in Alfalfa", which presents valuable research with comprehensive content and well-designed figures. However, the manuscript suffers from significant issues in writing quality and scientific rigor. The authors are advised to carefully review the entire manuscript. Below are some specific concerns:

The criteria used for identifying MsJAZ genes appear overly stringent. The study only includes genes containing both complete TIFY and Jas domains. However, in many plant species, JAZ proteins with incomplete or variant domains have also been reported and may still participate in JA signaling. Therefore, it is recommended that the authors consider including such variant forms to provide a more comprehensive overview of the MsJAZ gene family in alfalfa. Alternatively, the potential impact of these strict criteria on the gene family size should at least be addressed in the discussion.

Lines 56–57:The sentence “Plant cells synthesize JA in their chloroplasts, peroxisomes, and cytosol” is inaccurate. JA biosynthesis starts in the chloroplast, but the final synthesis of JA does not occur in the chloroplast. Please revise this statement accordingly.

Line 76 Glycine max should be italicized. Similarly, line 73 Cmjaz1 should use proper capitalization. The manuscript contains numerous Latin names and gene symbols that are not formatted correctly. The authors should carefully proofread and standardize nomenclature throughout.

Figure 3:The figure lacks legend and does not appear to be referenced in the main text.

Line 198: The term "covariance" is misused in this context. The authors likely mean "synteny" or "collinearity." Please confirm and revise the terminology. Also, Figure 4B is not clearly cited or discussed in the text.

Table 2:The Ks-related content is not explained or discussed in the manuscript.

Line 214:If the heatmap is based on RT-qPCR data, then using a heatmap is not appropriate as it omits error bars and statistical comparisons. Comparisons without significance testing are not suitable. Please revise the visualization method and include additional statistical analysis.

Lines 225–226: It is unclear what the correlations refer to. Correlation between what variables? Also, the phrase “significant direct correlation” is vague.

Figure 5A: If this figure is based on RT-qPCR, how were the corresponding values transformed?

Lines 252–253: The statistical method used for significance testing should be explicitly stated.

Figure 7: As with previous figures, if based on qPCR, important information is lost, including error bars. Furthermore, comparisons must be supported by proper statistical testing.

Lines 288–289: The sentence "This suggests that MsJAZ4 and MsJAZ7 encode nuclear and plasma membrane proteins" is incorrect. P The statement should be corrected to localization, not encoding.

Line 303: The figure is missing proper labeling and caption. Additionally, based on convention, Dilution should be arranged from low to high (left to right), which is currently reversed in the figure.

Lines 400–407: The abrupt mention of Cr (chromium) lacks context and relevance to the main theme of the manuscript. Please clarify its significance or remove it.

Lines 430–431:The statement “We deleted the gene containing the typical functional domain” ? What does that mean? How did the author make the selection?

Lines 438–439:The methods refers to AlphaFold predictions. Does the author's paper involve the research of Al phaFold?

Lines 464–465: The Ka/Ks analysis are mentioned but not discussed at all in the main text.

Line 474: What did the authors use to determine the selected concentrations and time points for treatments? Refs? or pre-experiment?

In summary, the manuscript contains a number of factual inaccuracies and major issues in expression and formatting. The authors should carefully revise the text to ensure scientific accuracy and clarity.

Author Response

Reviewer 1

Comments 1: Lines 56–57:The sentence “Plant cells synthesize JA in their chloroplasts, peroxisomes, and cytosol” is inaccurate. JA biosynthesis starts in the chloroplast, but the final synthesis of JA does not occur in the chloroplast. Please revise this statement accordingly.

Response1:Thanks for your suggestion. We have amended the statement in lines 56-57: ‘Plant cells synthesise JA in chloroplasts, peroxisomes and cytoplasm’.

Comments 2: Line 76 Glycine max should be italicized. Similarly, line 73 Cmjaz1 should use proper capitalization. The manuscript contains numerous Latin names and gene symbols that are not formatted correctly. The authors should carefully proofread and standardize nomenclature throughout.

Response 2:Thanks for your suggestion. We have italicised line 76 Glycine max. Similarly, line 73 Cmjaz1 has been changed to CmJAZ1. Many Latin names and gene symbols in the manuscript have been changed.

Comments 3: Figure 3:The figure lacks legend and does not appear to be referenced in the main text.

Response 3:Thanks for your suggestion. Figure 3: Legend has been represented in the figure and figure III is cited in the text. We have added the figure notes. It has been labelled in red in the text.

Comments 4: Line 198: The term "covariance" is misused in this context. The authors likely mean "synteny" or "collinearity." Please confirm and revise the terminology. Also, Figure 4B is not clearly cited or discussed in the text.

Response 4:Thanks for your suggestion. We have changed "covariance" to"Collinearity." It is marked in red. We refer to the collinearity discussion in the discussion section,There is a lack of research on the function of the JAZ gene across species. Therefore, there are no direct literature citations.

Comments 5: Table 2:The Ks-related content is not explained or discussed in the manuscript.

Response 5:Thanks for your suggestion.We have added a discussion of Ka/Ks in the Discussion section. 

Comments 6: Line 214:If the heat map is based on RT-qPCR data, then using a heat map is not appropriate as it omits error bars and statistical comparisons. Comparisons without significance testing are not suitable. Please revise the visualization method and include additional statistical analysis.

Response 6: Thanks for your suggestion. We represented gene tissue expression in heat map and analyzed significance using ANOVA. and supplemented in the Appendix by taking log2 for relative expression and visualization it used heat maps. And the results of the significance analyses are added in Annex 6.

Comments 7: Lines 225–226: It is unclear what the correlations refer to. Correlation between what variables? Also, the phrase “significant direct correlation” is vague.

Response 7: Thanks for your suggestion. Correlation refers to the correlation between the expression of different MsJAZ genes. From Figure 5B, there was a significant correlation in the expression of a few genes.

Comments 8: Figure 5A: If this figure is based on RT-qPCR, how were the corresponding values transformed?

Response 8: Thank you for your suggestion. We analysed the significance by ANOVA, log-transformed the means, and used heatmaps to represent the expression of gene tissues. We also added ANOVA significance and log-transformed data in the Appendix.

Comments 9: Lines 252–253: The statistical method used for significance testing should be explicitly stated.

Response9:Thanks for your suggestion. We have added tests of significance with analysis of variance (ANOVA) in lines 253-254 of the text, the same as below.

Comments 10: Figure 7: As with previous figures, if based on qPCR, important information is lost, including error bars. Furthermore, comparisons must be supported by proper statistical testing.

Response10: Thanks for your suggestion. Gene expression data The qPCR-based analysis of variance (ANOVA) of relative expression was followed by a transformation of the mean using log2 fold change, which is a more intuitive representation of gene up- or down-regulation.  We used ANOVA to compare the data for significance and added in the Appendix.

Comments 11: Lines 288–289: The sentence "This suggests that MsJAZ4 and MsJAZ7 encode nuclear and plasma membrane proteins" is incorrect. P The statement should be corrected to localization, not encoding.

Response11: Thanks for your suggestion. We have amended lines 288-289 of the text ‘This suggests that MsJAZ4 and MsJAZ7 encode nuclear and plasma membrane proteins’ to ‘This suggests that MsJAZ4 and MsJAZ7 proteins were localized to the nucleus and the cell membrane’.

Comments 12: Line 303: The figure is missing proper labeling and caption. Additionally, based on convention, Dilution should be arranged from low to high (left to right), which is currently reversed in the figure.

Response 12:Thanks for your suggestion.We have added to line 303 of the text, where the figure lacks proper labelling and description. In addition, arrange the dilutions in the image from highest to lowest (left to right).

Comments 13: Lines 400–407: The abrupt mention of Cr (chromium) lacks context and relevance to the main theme of the manuscript. Please clarify its significance or remove it.

Response13: Thanks for your suggestion. We have clarified the significance of the study of the heavy metal chromium, which is supplemented in the results section 2.7 in the text. The labelling is done in red font.

Comments 14: Lines 430–431:The statement “We deleted the gene containing the typical functional domain” ? What does that mean? How did the author make the selection?

Response 14: Thanks for your suggestion. I'm very sorry. We inadvertently misrepresented this. What we wanted to say was: Delete genes that do not have a typical structural domain. The original statement has been corrected. It is marked in red.

Comments 15: Lines 438–439: The methods refers to AlphaFold predictions. Does the author's paper involve the research of Al phaFold?

Response 15: Thanks for your comments. We only predicted the MsJAZ protein structure used AlphaFold software and did not involve AlphaFold studies.

Comments 16: Lines 464–465: The Ka/Ks analysis are mentioned but not discussed at all in the main text.

Response16:Thanks for your suggestion. We have added a discussion of Ka/Ks in the Discussion section. and labeled in red font.

Comments 17: Line 474: What did the authors use to determine the selected concentrations and time points for treatments? Refs? or pre-experiment?
Response17:Thanks for your comments.The 20% PEG-6000, 200 mM NaCl, 100 μM JA, and treatment time nodes used in the paper were obtained from references, and 300 μM CrCl₃-6Hâ‚‚O was the concentration screened in pre-laboratory experiments.

Reviewer 2 Report

Comments and Suggestions for Authors

The manuscript has a scientific contribution but there are some queries that need to be addressed:

  1. Line 30: Please add summary sentence highlighting potential future directions or practical implications from the research finding.
  2. Line 48: add reference.
  3. Line 69: add reference.
  4. The introduction is too lengthy and it should follow a logical structure of "posing the problem, analyzing the problem, and solving the problem.
  5. Line 124: The authors clustered Alfalfa JAZ genes with those from Arabidopsis, Zea mays, and Oryza sativa. Do they find the same clade of genes shows similar phytohormone response in these species.
  6. Line 264: Whether any common co-expressing genes was observed under cadmium and jasmonic acid stress treatments.
  7. There are many repeats in the results and discussion sections of the manuscript, so that the discussion section needs to be reorganized, also it lacks depth.
  8. The methods are well described overall and is relatively easy to follow.

Author Response

Reviewer 2

The manuscript has a scientific contribution but there are some queries that need to be addressed:

Comments 1: Line 30: Please add summary sentence highlighting potential future directions or practical implications from the research finding.

Response 1:Thanks for your suggestion. We have made a change in line 30 of the summary section.

Comments 2: Line 48: add reference.

Response 2:Thank you for your suggestion. We have added references to relevant studies on line 48 of the Received Manuscript and marked them in red font.

Comments 3: Line 69: add reference.

Response 3:Thank you for your suggestion.We have added references to relevant studies on line 69 of the Received Manuscript and marked them in red font.

Comments 4: The introduction is too lengthy and it should follow a logical structure of "posing the problem, analyzing the problem, and solving the problem.

Response 4:Thank you for your suggestion.

Comments 5: Line 124: The authors clustered Alfalfa JAZ genes with those from Arabidopsis, Zea mays, and Oryza sativa. Do they find the same clade of genes shows similar phytohormone response in these species.

Response 5:Thank you for your suggestion. We explain in the discussion section.

Comments 6: Line 264: Whether any common co-expressing genes was observed under cadmium and jasmonic acid stress treatments.

Response 6:Thank you for your suggestion. According to the experimental results, we found that the expression of MsJAZ4/7/9 was significantly elevated under cadmium and jasmonic acid treatments, and the expression of MsJAZ4 gene reached the maximum at 3 h after the treatment, and the expression of MsJAZ7 and MsJAZ9 genes reached the maximum at 12 h after the treatment. It has been noted in the manuscript and labeled in red.

Comments 7: There are many repeats in the results and discussion sections of the manuscript, so that the discussion section needs to be reorganized, also it lacks depth.

Response 7:Thank you for your comments.Our section on results and discussion needs to be reorganised.

Comments 8: The methods are well described overall and is relatively easy to follow.

Response 8:Thank you for your comments.

Reviewer 3 Report

Comments and Suggestions for Authors

I have not any coments. In the section discution in row number 17 first sentence is wrong write. I suggestion thet is sentence  "lucerna" changed in sentence Alfalfa or Medicago sativa.

Author Response

Reviewer 3

Comments 1: I have not any comments. In the section discution in row number 17 first sentence is wrong write. I suggestion that is sentence "lucerne" changed in sentence Alfalfa or Medicago sativa.

Response 1:Thanks for your suggestion. We have changed “lucerne” to “alfalfa” in the discussion section, line 17. Mark it in red.

Round 2

Reviewer 1 Report

Comments and Suggestions for Authors

The authors have addressed most of my concerns; however, I am confused by the discrepancy in the reported treatments. The abstract refers to cadmium (Cd), and line 269 mentions "responses to Cadmium and MeJA," while the text later discusses chromium (Cr), with line 506 specifying "300 µM CrCl₃•6Hâ‚‚O for different treatments." Cd and Cr are not the same compound. Could the authors please clarify whether the treatment used was Cd or Cr? If there was a mistaken use of "Cr" (chromium) instead of "Cd" (cadmium), or vice versa, the authors must carefully review and revise all instances throughout the manuscript where the metal is referenced.

Author Response

Reviewer 1

Comments 1: The authors have addressed most of my concerns; however, I am confused by the discrepancy in the reported treatments. The abstract refers to cadmium (Cd), and line 269 mentions "responses to Cadmium and MeJA," while the text later discusses chromium (Cr), with line 506 specifying "300 µM CrCl₃•6Hâ‚‚O for different treatments." Cd and Cr are not the same compound. Could the authors please clarify whether the treatment used was Cd or Cr? If there was a mistaken use of "Cr" (chromium) instead of "Cd" (cadmium), or vice versa, the authors must carefully review and revise all instances throughout the manuscript where the metal is referenced.

Response 2: I'm very sorry. Due to our negligence, cadmium and chromium were confused and misused. We have corrected the article where chromium is concerned and marked it in red.